# Detection of Canopy Chlorophyll Content of Corn Based on Continuous Wavelet Transform Analysis

**Junyi Zhang [1,2], Hong Sun [1], Dehua Gao [1], Lang Qiao [1], Ning Liu [1], Minzan Li [1,\*] and Yao Zhang [1]**

[1]   Key Laboratory of Modern Precision Agriculture System Integration Research, Ministry of Education, China Agricultural University, Beijing 100083, China; junyizh@cau.edu.cn (J.Z.); sunhong@cau.edu.cn (H.S.); dehua_gao@cau.edu.cn (D.G.); b20193080667@cau.edu.cn (L.Q.); ningliu@cau.edu.cn (N.L.); zhangyao@cau.edu.cn (Y.Z.)

[2]   College of Energy and Intelligence Engineering, Henan University of Animal Husbandry and Economy, Zhengzhou 450046, China

\*   Correspondence: limz@cau.edu.cn

**Abstract:** The content of chlorophyll, an important substance for photosynthesis in plants, is an important index used to characterize the photosynthetic rate and nutrient grade of plants. The real-time rapid acquisition of crop chlorophyll content is of great significance for guiding fine management and differentiated fertilization in the field. This study used the method of continuous wavelet transform (CWT) to process the collected visible and near-infrared spectra of a corn canopy. This task was conducted to extract the valuable information in the spectral data and improve the sensitivity of chlorophyll content assessment. First, a Savitzky–Golay filter and standard normal variable processing were applied to the spectral data to eliminate the influence of random noise and limit drift on spectral reflectance. Second, CWT was performed on the spectral reflection curve with 10 frequency scales to obtain the wavelet energy coefficient of the spectral data. The characteristic bands related to chlorophyll content in the spectral data and the wavelet energy coefficients were screened using the maximum correlation coefficient and the local correlation coefficient extrema, respectively. A partial least-square regression model was established. Results showed that the characteristic bands selected via local correlation coefficient extrema in a wavelet energy coefficient created a detection model with optimal accuracy. The determination coefficient ($R_c{}^2$) of the calibration set was 0.7856, and the root-mean-square error (*RMSE*) of the calibration set (*RMSEC*) was 3.0408. The determination coefficient ($R_v{}^2$) of the validation set is was 0.7364, and the *RMSE* of the validation set (*RMSEV*) was 3.3032. Continuous wavelet transform is a process of data dimension enhancement which can effectively extract the sensitive variables from spectral datasets and improve the detection accuracy of models.

**Keywords:** canopy spectra; chlorophyll content; continuous wavelet transform (CWT); correlation coefficient; partial least square regression (PLSR)

## 1. Introduction

Corn, one of the major food crops in the world, provides an important guarantee of food security and economic development. Proper nitrogen application is one of the keys to a good harvest of corn [1,2]. A number of studies have shown that the leaf chlorophyll content (LCC) can be used to predict the nitrogen requirement of crops [3,4]. Chlorophyll content is an important indicator of crop photosynthesis ability and nutrition level. Variable fertilization can be achieved using nitrogen fertilizers according to different chlorophyll contents through accurate monitoring of the chlorophyll content of corn leaves [5,6]. Appropriate fertilization can ensure that crops receive adequate nitrogen

and avoid soil and water pollution caused by excessive fertilization [7]. This mechanism is the key to improving the photosynthetic performance of crops, thereby regulating the growth and development of corn and increasing the input–output ratio of corn fertilizers [8]. The topic of detecting chlorophyll content of corn is one of the active areas in field management research today. Thus, this study aimed to detect the chlorophyll content in field crops to evaluate the growth status and providing guidance for fertilization.

The traditional chlorophyll detection method is an analytical chemical method, which has high precision. However, the process is complex, time-consuming, and may damage crops. This method cannot meet the requirements of rapid and nondestructive testing on site. Spectral analytical technology has been widely used in qualitative and quantitative analysis of the physicochemical parameters of farmland crops because of its fast, nondestructive, and nonpolluting characteristics. Kapp-Junior et al. [9] developed a novel regression model able to produce a prescription of the required nitrogen (N) for maize by combining spectral reflectance data and agronomic efficiency. Lu et al. [10] used hyperspectral techniques to analyze the vertical distribution of nitrogen in corn. Zhang et al. [11] used leaf characteristic spectra to forecast apple sugar content. These studies highlight the feasibility and efficiency of evaluating crop nutrients via spectroscopy. Therefore, rapid detection of the chlorophyll content was conducted in the present research by using hyperspectral technology during the growth stages.

Most current studies on the detection of chlorophyll content via spectral analysis have focused on the exploration of spectral characteristics to quantitate the intensity and position of molecular absorption or reflection [12,13]. The two types of methods to quantify the spectral absorption and reflectance of specific matters include multivariate statistical analysis and region positioning calculation. First, multivariate statistical methods are used to select and enhance the parameters of spectral reflectance, derivative spectrum, and vegetation index using maximum correlation coefficient analysis. Liang et al. [14] compared fifty hyperspectral vegetation indices, such as the photochemical reflectance index and canopy chlorophyll index, to identify the most appropriate vegetation indices for crop LCC and canopy chlorophyll content (CCC) inversion. Xu et al. [15] used simulated datasets from the PROSAIL model to establish a 2D-matrix-based relationship between leaf chlorophyll and red-edge relative indices ($RERI_{(705nm)}$ and $RERI_{(783nm)}$). The leaf chlorophyll content can be retrieved using the two vegetation indices from observations on the basis of the matrix. Neto et al. [16] created a sunflower leaf chlorophyll model with the spectral reflectance in the band of 500–1039 nm by using partial least-squares regression (PLSR). Rei et al. [17] used two methods, namely machine-learning algorithms and the inversion of a radiative transfer model, to detect the LCC of tea. Second, the characteristic spectral positions show changes with the local correlation extreme values, which generally include the red edge and green peak. Li et al. [18] used spectral reflectance to construct red-edge spectral parameters and newly developed red-edge region parameters to detect the chlorophyll content in rapeseed leaves. Sun et al. [19] indicated that the blue edge, red valley, and eight other spectral parameters could be used to reflect the chlorophyll content of potato crops. Zheng et al. [20] developed a model of chlorophyll content in potato leaves on the basis of the red-edge location. The mentioned characteristic parameters selected by maximum or local extreme correlation have been widely used as sensitive spectral variables for detecting the chlorophyll content.

However, the canopy reflectance spectrum and sensitive bands of crops are easily affected by external interferences of dynamically changing soil background, vegetation canopy geometry, and atmospheric conditions during the growth periods [21–23]. Numerous studies have attempted to improve the detection models of chlorophyll content by eliminating the irrelevant and noise information of the spectral data [24,25]. With regard to the ideas and principles of radiation transmission, the combination of the PROSPECT leaf optical property model and SAIL (Scattering by arbitrarily inclined leaves) model, also referred to as PROSAIL, has been used to develop methods for retrieval of vegetation biophysical properties. Mridha et al. [26] used the broadband canopy radiation transfer model PROSAIL to invert the leaf area index (LAI), LCC, CCC, and leaf equivalent water thickness

of the biophysical variables in soybeans. Botha et al. [27] evaluated the ability of the PROSAIL canopy-level reflectance model to detect LCC of spring wheat (*Triticum aestivum* L.) during the growth stages between pretillering (Zadoks growth stage (ZGS15)) and booting (ZGS50). Lunagaria et al. discussed the spectral sensitivity of crop canopy parameters using a theoretical model. The results indicated that the reflectance in the visible range was important for chlorophyll retrieval. Reflectance in the near-infrared range has importance for retrieval of leaf inclination angle, dry matter, and LAI. Accordingly, the influencing factors are difficult to reduce, and the modeling results are challenging to improve because of the external and internal interferences affecting the field canopy spectrum [28]. The primary concern of this research area is to overcome such challenges to detect the chlorophyll content by hyperspectral technology during the growth stages.

Although scholars have tried to use preprocessing methods (such as continuum removal, first-order differentiation, and high-pass filtering) to eliminate the noise and enhance the characteristic signals caused by certain factors (such as sample background and stray light), challenges and problems in effectively removing the interference signals, especially random and low-frequency signals, exist during dynamic growth periods [29–31]. We addressed the primary concern using continuous wavelet transform (CWT) to overcome such problems. CWT, with its rich wavelet base function, multiresolution analysis, time-frequency localization, and other advantages, has received increasing attention in image and signal analysis, decomposition, compression, and denoising [32–37]. This method can effectively separate low-frequency signals from high-frequency signals and extract the weak information from the spectral signal. Chen et al. [38] studied the CWT, taking 265 leaves of 47 plants as the sample spectrum and effectively inverting the water content in the sample, with a high precision of up to 75%. Li et al. proposed a new technique (WREP) to extract red-edge positions (REPs) on the basis of the application of CWT to the reflectance spectra. The results demonstrated that WREP obtained the best detection accuracy for LCC and CCC compared with the traditional techniques. High scales of wavelet decomposition were favorable for the detection of CCC and low scales for the detection of LCC [39]. These studies highlight that the CWT can be used to improve the modeling results of LCC detections. However, great uncertainty still exists regarding the effects of this method to help improve LCC detection during growth periods in which the spectral characteristics are dynamically changed and influenced by soil background, vegetation canopy geometry, and atmospheric conditions. Similar to spectral wavelengths, whether to use the local correlation extremum method or the maximum correlation coefficient method to select sensitive wavelet features is worth discussing. Thus, this study aimed to clarify and create a model to monitor the CCC of corn on the basis of CWT during the growth stages.

This study focused on the relationship between LCC and corn canopy spectral reflectance to propose an efficient method to evaluate the chlorophyll content of corn. The main aims of this study were as follows: (1) to comparative analyze the advantages of maximum correlation value and local correlation extreme value in selecting feature variables; (2) to use CWT to decompose the original spectral data and extract the weak information in the spectrum to detect the chlorophyll content.

## 2. Materials and Methods

### 2.1. Experiments and Materials

The experiments were conducted in Hengshui City, Hebei Province, China. Seventy-two sampling areas were present in the test field, as shown in Figure 1, with six fertilization levels to ensure the gradient of chlorophyll content. The nitrogen fertilizer was pure nitrogen, and the phosphorus fertilizer was $P_2O_5$. The spectral data were collected in three growth periods on the basis of the growth time and conditions, namely, G1 (6 leaf stage), G2 (9 leaf stage), and G3 (12 leaf stage). During growth periods, one leaf sample was collected in each sampling area, so that two hundred and sixteen leaf samples were collected.

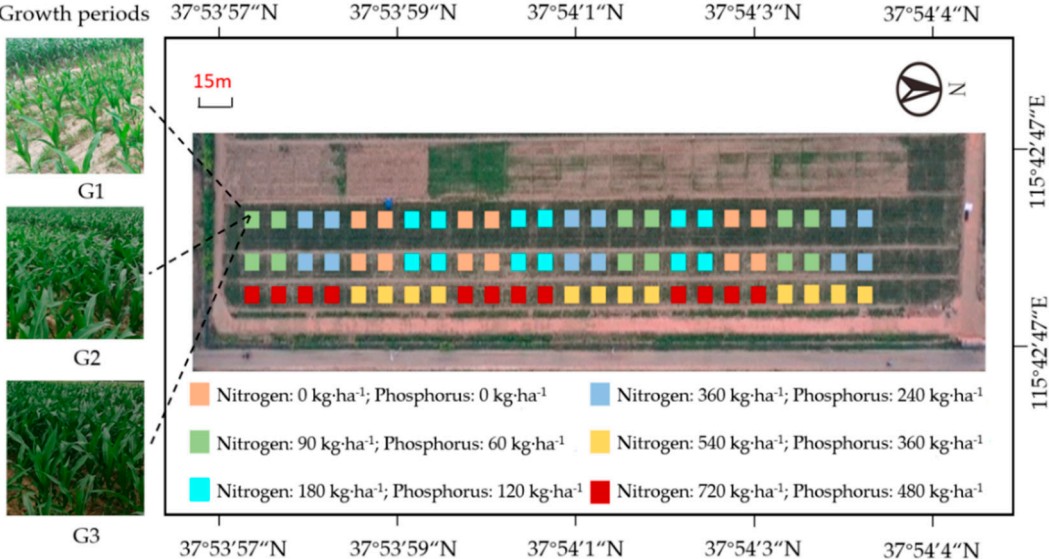

**Figure 1.** Locations and treatments of the experiment.

The overall process of analyzing reflection spectrum data and chlorophyll content is shown in Figure 2; this mainly included the collection of spectrum data, preprocessing of the spectral data, selection of the characteristic variables, and establishment of a detection model for the chlorophyll content. The selection of the characteristic variables was conducted on the basis of comparison of the characteristic wavelengths and wavelet features selected by maximum correlation coefficients and local extrema of the correlation coefficients.

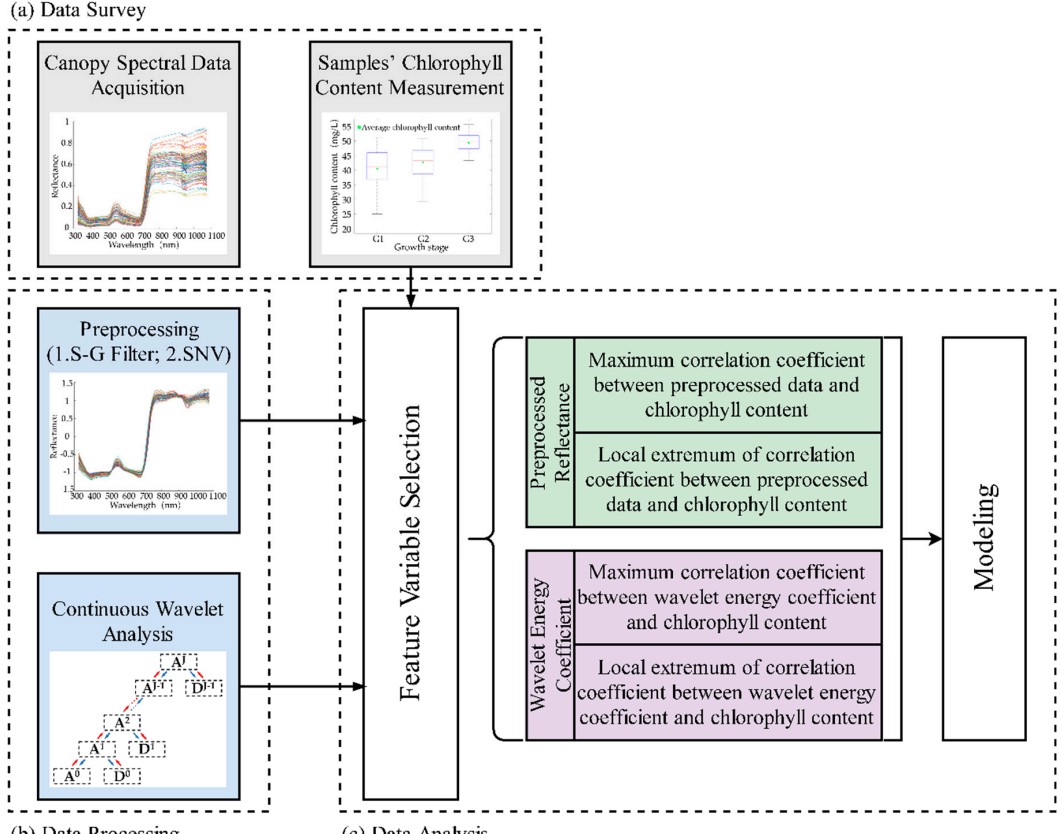

**Figure 2.** Data analysis flow chart.

## 2.2. Field Spectrum Data Collection and Chlorophyll Content Measurement

The ASD FieldSpec® HandHeld 2 was used in the field to measure the canopy reflectance of corn. This tool is a hand-held spectrometer with a wavelength range of 325–1075 nm, wavelength accuracy of 1 nm, and spectral resolution of <3.0 nm at 700 nm [40]. Sample leaves were randomly chosen in each sample area to measure the spectral reflectance. Sample leaves were then sealed for subsequent chlorophyll extraction experiment. Spectrum reflectance data were collected three times above the leaf during the spectrum measurement. The averaged reflectance was taken as an original spectral datum.

The chlorophyll content of the sample leaves was measured in the laboratory via SHIMADZU UV2450 spectrophotometry. The spectrophotometry measurement wavelength range was 190–900 nm and the band width was 0.1–5 nm. The spectral resolution was 0.1 nm and the stray light was lower than 0.015%. The main stems of corn leaves were removed, and the leaves were shredded and evenly mixed. Approximately 0.4 g crushed leaf samples were soaked in 25 mL acetone and absolute ethanol mixture for 24 hours, and the mixture ratio was 2:1. During soaking, the mixed solutions were shaken three times to accelerate the chlorophyll extraction. The absorbances of extract solution at 645 and 663 nm were measured with a UV spectrophotometer. The concentrations of chlorophyll *a* ($Chl_a$) and chlorophyll *b* ($Chl_b$) were calculated using the following equations:

$$Chl_a = 12.72 \times A_{663} - 2.59 \times A_{645}, \tag{1}$$

$$Chl_b = 22.88 \times A_{645} - 4.67 \times A_{663}, \tag{2}$$

$$Chl_T = Chl_a\left(\text{mg L}^{-1}\right) + Chl_b\left(\text{mg L}^{-1}\right), \tag{3}$$

where $A_{645}$ and $A_{633}$ are the absorbances of the extract solution at 645 and 663 nm, respectively, and $Chl_T$ is the total chlorophyll [41].

## 2.3. Spectrum Data Preprocessing

The corn canopy spectrum collected in the field environment contained noise information due to the uneven surface of the sample, random noise, different optical paths, and light scattering. First, a Savitzky–Golay (S-G) filter was used to smoothen the reflection spectrum, and the smoothing window was set to 13 [42]. The S-G filter is based on the principle of least squares. Multiple fitting was performed to the original signal in the correction window and the final conversion result was calculated by the multiplicity of the fitting. Using *m* (*m* is odd) continuous wavelength points as the window, the data points inside the smooth window were fitted by *p*-order polynomial function, and the polynomial equation combination was obtained. The smoothing coefficient was obtained using least-square fitting, and the corrected spectral value of the center point of the window was calculated. By successively moving the position of the smoothing window and repeating the above polynomial fitting steps, the spectra after S-G filtering were obtained.

Second, the standard normal variable (SNV) method was used to process the smoothed spectral curve to reduce the influence of the scattering effect [43]. Standard normal variable correction is often used to eliminate the effects of different particle sizes, surface scattering, and optical path differences in NIR diffuse reflectance spectra. The SNV correction of sample spectra were independent of each other and did not involve the spectral information related to the sample set. First, the sample spectrum was centralized, which means that the mean value of spectral reflectance of each spectral data was subtracted from the sample. The standard deviation of the sample reflectance was then used to scale up. After SNV correction, the spectral mean of each sample became 0 and the variance became 1. The SNV correction spectrum of sample *j* is as follows:

$$A_{j,SNV} = \left(A_j - \overline{A_j}\right)/\sigma_j, \tag{4}$$

where $\overline{A_j}$ is the mean value of the spectrum of sample $j$ and $\sigma_j$ is the standard deviation of the spectrum of sample $j$.

Wavelet analysis is one of the potential technologies used in the extraction of weak hyperspectral information. Wavelet transform is a function combination that decomposes a complex signal into simple subsignal components. The spectral signal can be decomposed into subsignals of different frequencies when applied to the analysis of crop spectral data. We effectively used the overall structural characteristics of spectral information and extracted the weak information hidden in the spectral signal. Moreover, we searched for the optimal combination of the subsignal components to detect the chlorophyll content of the crop canopy.

### 2.4. Sample-Set Division Algorithm According to Sample-Set Partitioning Based on the Joint X–Y Distance (SPXY)

In this study, the SPXY algorithm proposed by Galvão et al. was used to divide the modeling and verification sets [44]. This approach is a method for dividing the sample set on the basis of the statistical perspective, and it comprehensively considers the difference between the spectrum and the property parameters to select the modeling set. The SPXY algorithm first calculates the Euclidean distance between the spectrum data of all samples using Equation (5). The algorithm then selects the two with the largest distance as the first two samples in the modeling set.

$$d_x(p,q) = \sqrt{\sum_i^I \left[x_p(i) - x_q(i)\right]^2}; \ p,q \in [1,N],$$ (5)

where $x_p(i)$ and $x_q(i)$ are the spectral parameters of samples $p$ and $q$ at $i$ wavelength, respectively; $I$ is the number of wavelengths in the spectrum; and $N$ is the number of samples.

The Euclidean distances between the remaining and selected samples were calculated. The sample with the next longest Euclidean distance was selected as the third sample in the modeling set. We repeated the above-mentioned steps until the number of selected samples was equal to the predetermined number.

The nature property factor $d_y(p,q)$ was considered as Equation (6) on the basis of the above-mentioned formula.

$$d_y(p,q) = \sqrt{\left(y_p - y_q\right)^2}; \ p,q \in [1,N] ,$$ (6)

where $y_p$ and $y_q$ are the property parameters of samples $p$ and $q$, respectively.

Variables $d_x(p,q)$ and $d_y(p,q)$ were divided by their maximum values in the dataset to ensure that the sample had the same weight in the spectral and property spaces. The standardized $xy$ distance formula was as follows:

$$d_{xy}(p,q) = \frac{d_x(p,q)}{max_{p,q\in[1,N]}d_x(p,q)} + \frac{d_y(p,q)}{max_{p,q\in[1,N]}d_y(p,q)}.$$ (7)

### 2.5. Spectrum Characteristic Variable Selection Method

2.5.1. Characteristic Variable Selection

(1)  Maximum Correlation Coefficient Method

The characteristic variables needed to be filtered to simplify the model and improve its accuracy. The correlation analytical method is widely used to select variables highly correlated with chlorophyll content on the basis of the maximum correlation coefficient method. However, the selection result showed multicollinearity between the adjacent wavelength variables [45].

(2)　　Local Extremum of Correlation Coefficient Method

The local extremum of the correlation coefficient method was proposed to improve the variable selection strategy of correlation analysis to solve multicollinearity between the adjacent wavelength variables: the correlation coefficient of spectral reflectance and chlorophyll content were calculated, and the correlation coefficient curve was drawn. Thereafter, the local extreme points of the correlation curve (the zero-crossing positions of the correlation first-derivative curve) were calculated as the chlorophyll characteristic wavelengths [46]. The maximum correlation wavelengths were also selected on the basis of the maximum correlation coefficient method for comparative analysis.

### 2.5.2. Continuous Wavelet Analysis

From the perspective of signal processing, wavelet analysis can be used to perform data analysis in the frequency and time domains and extract available information from the signal [47]. The reflection spectrum analysis is highly similar to electronic signal analysis. Accordingly, CWT can be used to decompose the reflection spectrum curve at different frequency scales to generate a series of wavelet energy coefficients. The CWT process is shown in Equation (8).

$$W_f(a,b) = \frac{1}{\sqrt{a}} f(\lambda) \psi(\frac{\lambda - b}{a}) d\lambda, \tag{8}$$

where $a$ is the frequency scale factor which is set to $2^n$ ($n = 1, 2, \ldots, 10$) gradients, and translation factor $b$ is the center wavelength of the mother wavelet function. The mother wavelet function $\psi(\lambda)$ uses the second-order Gaussian function. $f(x)$ is a 1D reflection spectrum, and the wavelet coefficient $W_f(a,b)$ (denoted as $WF_{a,b}$) is 2D data, including frequency scale (1, 2, $\ldots$, 10) and wavelength (325–1075 nm).

Correlation analysis of the wavelet energy coefficient and chlorophyll content was performed. The local extreme value of the correlation was calculated as the sensitive wavelet feature. The wavelet coefficient with the highest correlation was selected for comparative analysis.

The extreme point was calculated using the peak function in MATLAB software. The parameters of the peaks' function were set as follows: the correlation extreme value peak distance of the spectral reflectance was set to 10, and the minimum peak value was set to 0.1; the correlation extreme value peak distance of the wavelet coefficient was set to 50, and the minimum peak value was set to 0.3.

### 2.6. Establishing Chlorophyll Content Detection Model Based on PLSR

PLSR gradually became a widely used modeling method in spectral analysis after it was proposed by Geladi in 1986 [48–50]. PLSR can solve the problems of autocorrelation and multicollinearity between variables on the basis of the method of principal component extraction. PLSR was used to perform principal component decomposition simultaneously on the spectral reflectance matrix and LCC matrix. During decomposition, PLSR correlated the spectral and chlorophyll content matrixes and established a linear regression model between the two to detect the chlorophyll content of corn leaves. The leave-one-out cross-validation (LOOCV) method was used for internal interactive verification, and the optimal number of characteristic variables was determined by root-mean-square error of cross-validation (RMSECV). The model evaluation indicators were the validation coefficient of validation set model ($R_v{}^2$) and the root-mean-square error of validation set (*RMSEV*).

## 3. Results

### 3.1. Analysis of Canopy Spectral Response of Corn during Growth Periods

The original reflectance spectrum of the corn canopy is shown in Figure 3a. The figure demonstrates serious noise-point information in the spectral curve. The noise point of the spectral curve was significantly reduced after the S-G filtering (Figure 3b). The scattering effect of the sample reflection

spectrum was significantly improved after the SNV correction (Figure 3c). The average spectral curves of three growth stages are shown in Figure 3d.

In general, 400–500 and 611–710 nm were two low-reflectance regions in the visible light band due to the strong absorption of blue and red light by leaf pigments. The absorption valleys appeared near 400 and 680 nm. Approximately 520–610 nm was a high-reflection area due to the strong reflection by the leaf pigments of green light. The reflection peak appeared near 550 nm. In the near-infrared region, the reflectance sharply increased from 711 nm to 760 nm due to the large cavity of the reflective surface in the spongy tissue structure of the mesophyll, thereby showing a "rapid climb" trend. The 761–1000 nm region was a strong reflection area, and the curve was close to horizontal, thereby showing a "high reflection platform". A weak absorption valley appeared around 970 nm due to the absorption of water.

Figure 3d demonstrates that the different growth periods varied in the four spectral ranges of 325–400, 401–700, 761–970, and 971–1075 nm. The spectral reflectance increased with the growth period in the ranges of 325–400 and 761–970 nm. The reflectance decreased with the growth period in the ranges of 401–700 and 971–1075 nm.

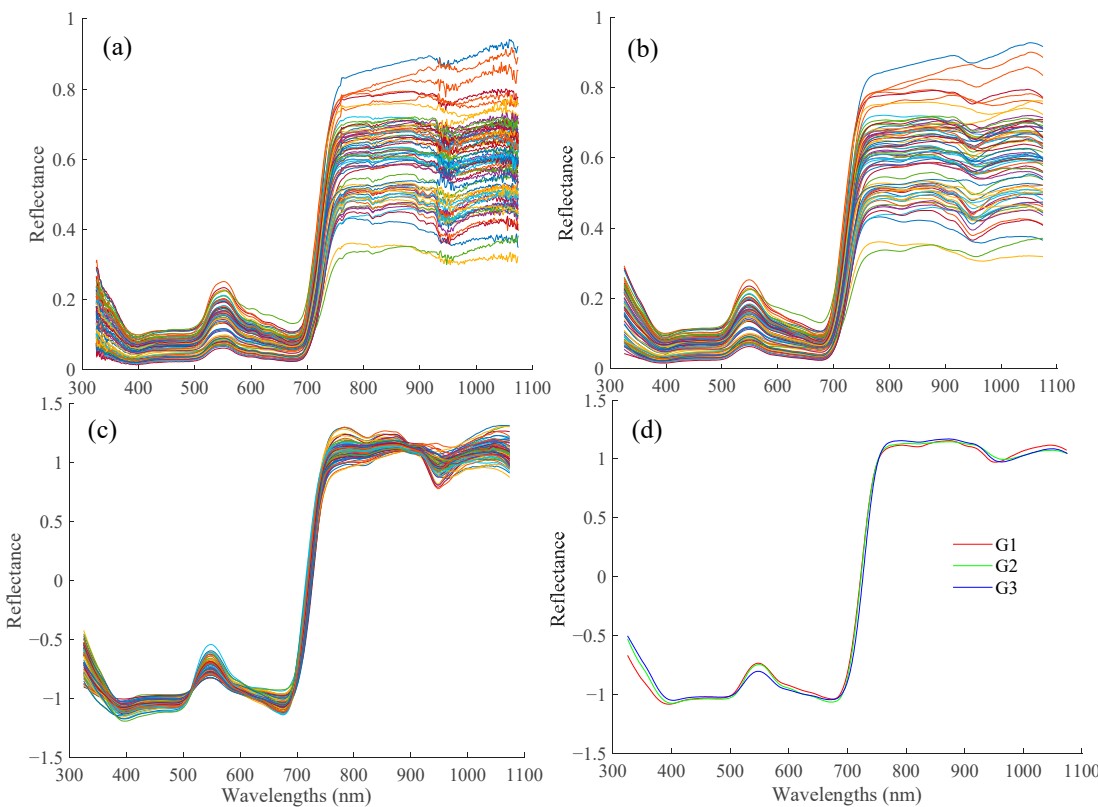

**Figure 3.** Corn canopy reflectance spectral curve. (**a**) Original canopy reflectance spectra; (**b**) canopy reflectance spectral after the S-G filtering; (**c**) canopy reflectance spectra after the S-G filtering and SNV; (**d**) average canopy spectra of three growth stages.

### 3.2. Statistical Analysis and Sample-Set Division

The trend of the average chlorophyll content with the growth period is shown in Figure 4, which demonstrated an increase from G1 to G3. From G1 to G3, the variation ranges of the chlorophyll content between samples gradually concentrated. The SPXY algorithm was used to divide the sample set according to the ratio of 2:1. The division result is shown in Table 1. One hundred and forty-four samples were included in the modeling set to establish a chlorophyll content detection model, and 72 samples in the verification set to test the performance of the detection model. The range of chlorophyll content of the samples in the modeling set was larger than that in the verification set. Thus, the sample set obtained by the SPXY algorithm was reasonable and was used for subsequent modeling.

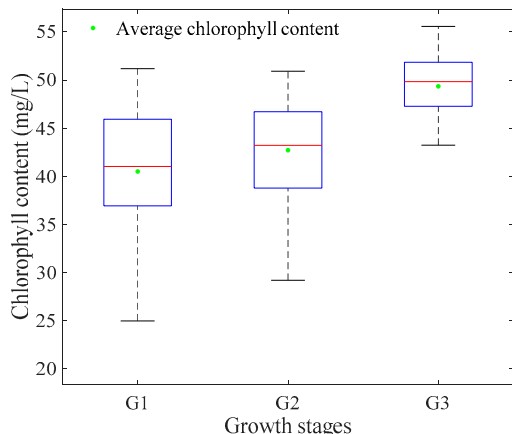

**Figure 4.** Statistical box line graph of chlorophyll content of the corn growth stages.

**Table 1.** Statistical results of the calibration set and validation set (%).

| Sample Set | Sample Size | Maximum | Minimum | Average | Standard Deviation |
|---|---|---|---|---|---|
| Total sample | 216 | 55.58 | 20.11 | 44.19 | 6.51 |
| Modeling set | 144 | 55.58 | 20.11 | 44.11 | 6.59 |
| Verification set | 72 | 55.22 | 25.31 | 44.35 | 6.38 |

*3.3. Correlation Analysis of the Chlorophyll Content and Spect/ral Reflectance*

3.3.1. Characteristic Wavelength Selection Based on the Maximum Correlation Coefficient Method

The correlation curve of the chlorophyll content and spectral reflectance is shown in Figure 5. The chlorophyll content was positively correlated with spectral reflectance in the blue (450–500 nm) and red (620–780 nm) regions, and negatively correlated in the green region. This result is consistent with the absorption characteristics of chlorophyll in visible light. The absolute value of the correlation coefficient between the chlorophyll content and the spectral reflectance was higher than 0.5 in the five bands, namely, 376–504, 518–596, 671–681, 698–746, and 880–913 nm. The correlation coefficient gradually decreased.

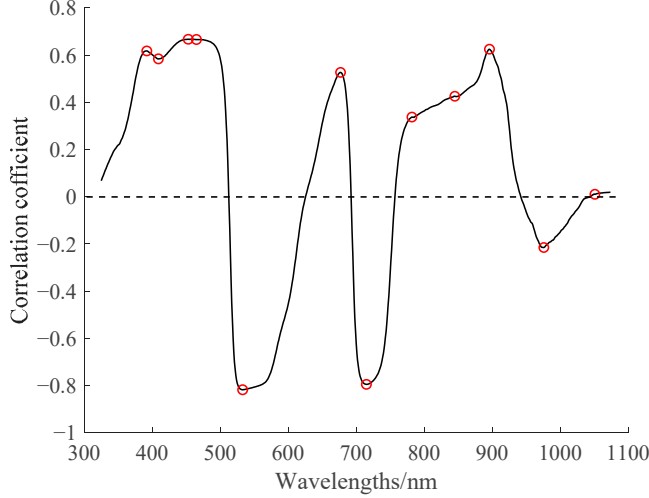

**Figure 5.** Correlation curve between the spectral reflectance and the chlorophyll content.

The top 15 wavelengths with the maximum correlations were selected as the chlorophyll-sensitive wavelengths (denoted as CA bands). The results were concentrated in the green light region of 529–543 nm. The CA bands were only divided into narrow bands. Accordingly, information redundancy may occur due to autocorrelation of information caused by adjacent narrow-band wavelengths.

### 3.3.2. Characteristic Wavelength Selection Based on the Local Extrema of the Correlation Coefficient

Twelve extreme points (denoted as CA peak bands) of correlation existed between the spectral reflectance and the chlorophyll content in five highly correlated bands. Approximately 392 and 409 nm were present in the violet region, 453 and 465 nm in the blue region, 533 nm in the green region, 677 nm in the red region, 715 and 782 nm in the red edge, and 845, 896, 976, and 1051 nm in the near-infrared region. The selected wavelengths are shown in Figure 4 with red circles.

### 3.4. Correlation Analysis of Chlorophyll Content and Wavelet Energy Coefficient

CWT was performed with 10 frequency scales on the spectral reflection curve. We calculated the correlation coefficients of the wavelet energy coefficient and chlorophyll content at each frequency scale. Thereafter, we took the absolute value of the correlation coefficient results and drew the distribution map of the correlation coefficients at different scales. The result is shown in Figure 6.

The wavelet energy coefficient bands with high correlation with chlorophyll content ($|r| > 0.5$) are shown in Table 2. Most of high-correlation wavelengths were concentrated in the visible light area of 325–700 nm. Selected numbers of wavelet energy bands were reduced with the scale increase. In the low-frequency scales from 1 to 4, the high-correlation bands had narrow wavelength ranges, more numbers, and clear division of intervals. In the mid- and high-frequency scales from 5 to 8, selected bands had wider wavelength ranges, less numbers, and ambiguous division of intervals. At the low-frequency scale, the high correlation bands of scale 1 and 2 were consistent, and the high correlation bands of scale 3 and 4 were consistent. With the increase of frequency scale, the correlation between the wavelet energy coefficient and chlorophyll decreased gradually from scale 5, and the boundary between the high-correlation band and the low-correlation band became ambiguous. It can also be seen from Figure 6 that as the frequency scale increased, the correlation between the wavelet energy coefficient and the chlorophyll content gradually decreased. No high-correlation wavelet energy coefficient exists on the high-frequency scale of 9–10. Therefore, the optimal features of wavelet energy should be selected on a scale of 1 to 5.

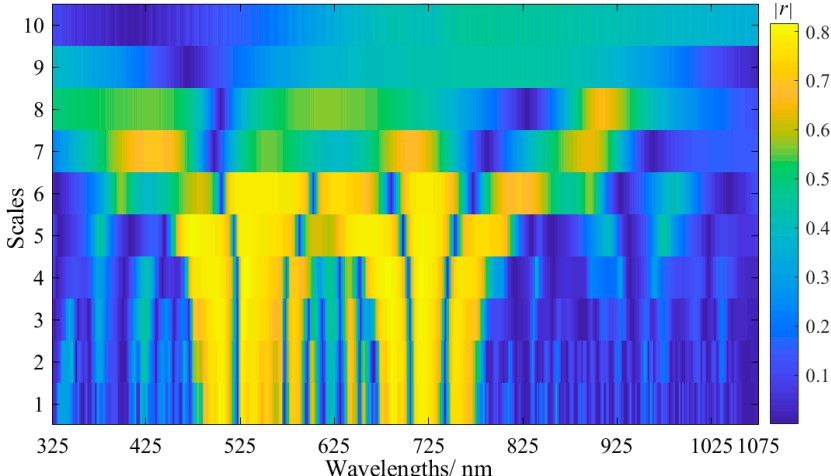

**Figure 6.** Distribution of the absolute values of the correlation coefficients between the wavelet energy coefficient and the chlorophyll content.

**Table 2.** Wavelet energy coefficient bands with high correlation with chlorophyll content ($|r| > 0.5$).

| Scale | Wavelengths |
|:---:|:---|
| 1 | 480–516 nm, 525–565 nm, 575–590 nm, 599–604 nm,635–645 nm, 671–700 nm, 710–740 nm, 745–775 nm. |
| 2 | 480–516 nm, 525–565 nm, 575–590 nm, 599–604 nm,635–645 nm, 671–700 nm, 710–740 nm, 745–775 nm. |
| 3 | 480–516 nm, 525–565 nm, 575–590 nm, 635–645 nm,671–700 nm, 710–740 nm, 745–775 nm. |
| 4 | 480–516 nm, 525–565 nm, 575–590 nm, 635–645 nm,671–700 nm, 710–740 nm, 745–775 nm. |
| 5 | 455–500 nm, 520–585 nm, 605–675 nm, 700–750 nm,758–811 nm. |
| 6 | 455–500 nm, 520–585 nm, 605–675 nm, 700–750 nm,788–858 nm, 886–901 nm. |
| 7 | 376–470 nm, 532–582 nm, 665–740 nm, 854–918 nm. |
| 8 | 326–466 nm, 555–695 nm, 883–944 nm. |

### 3.4.1. Sensitive Wavelet Feature Selection Based on the Maximum Correlation Coefficient

Fifty wavelet energy coefficients with high correlations were selected as the chlorophyll-sensitive wavelet features (denoted as CA-WFs). These result are shown in Table 3. The absolute values of the correlation coefficients of CA-WFs and chlorophyll content were all higher than 0.8.

**Table 3.** Location and frequency scale parameters of the chlorophyll-sensitive wavelet features (CA-WFs) with $|r|$ higher than 0.8.

| Wavelet Feature | Scale | Wavelengths |
|:---:|:---:|:---:|
| CA-WF1 | 1 | 504–505 (2) |
| CA-WF2 | 2 | 527 |
| CA-WF3 | 2 | 687 |
| CA-WF4 | 3 | 527–529 (3) |
| CA-WF5 | 3 | 685–688 (4) |
| CA-WF6 | 3 | 716–718 (3) |
| CA-WF7 | 4 | 528–534 (7) |
| CA-WF8 | 4 | 679–685 (7) |
| CA-WF9 | 4 | 715–720 (6) |
| CA-WF10 | 5 | 472–483 (12) |
| CA-WF11 | 5 | 717–720 (4) |

### 3.4.2. Sensitive Wavelet Feature Selection Based on the Local Extrema of the Correlation Coefficient

Fifty-five chlorophyll-sensitive wavelet energy coefficients were selected as sensitive wavelet features (denoted as CA peak WFs) on the basis of the extreme values of the correlation coefficient. The absolute values of the correlation coefficients of CA peak WFs and chlorophyll content were all higher than 0.5. The distribution of CA peak WFs at different frequency scales is shown in Figure 7.

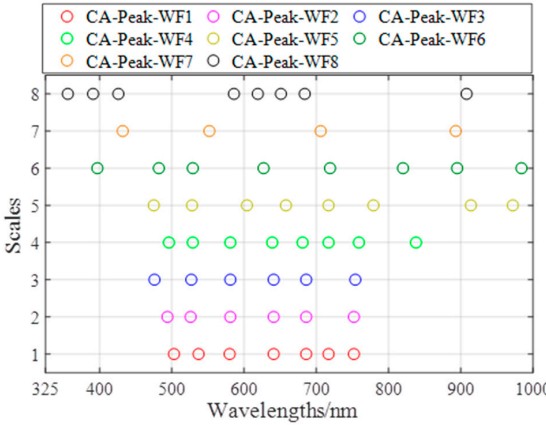

**Figure 7.** Result of the sensitive wavelet feature selection based on the local correlation coefficients.

The wavelength position analysis indicated that all sensitive wavelet variables of the CA Peak WF1, CA Peak WF2, and CA Peak WF3 were distributed in the visible light region, reflecting the leaf pigment information. CA Peak WF4 contained eight sensitive variables, seven of which were located in the visible light region, and another variable was located at 839 nm. CA Peak WF5 contained eight sensitive variables, six of which were located in the visible light region, and the other two variables were located at 915 and 973 nm. The 915 nm wavelength can reflect $CH_2$ methylene groups, and 973 nm can present the leaf moisture information. CA Peak WF6 contained eight sensitive variables, five of which were located in the visible light region, and the other three variable positions were 821, 896, and 985 nm. CA Peak WF7 and CA Peak WF8 contained four and eight variables, respectively; each of them had one sensitive variable located in the near-infrared region at 894 and 909 nm. The remaining sensitive variables were located in the visible region.

### 3.5. Establishment of Chlorophyll Content Detection Model with PLSR

The PLSR algorithm was used to establish a chlorophyll content detection model on the basis of the spectral characteristic variables of the CA bands, CA peak bands, CA-WFs, and CA peak WFs. Both models used LOOCV for internal cross-validation to eliminate the influence of spectral information redundancy and multicollinearity on the model accuracy. The modeling results are shown in Table 4 and the verification results are shown in Figure 8. The comparison of the four detection models showed that the PLSR chlorophyll content detection model based on CA peak WFs had the optimal performance. The decision coefficient ($R_c^2$) of the modeling set was 0.7856, the *RMSEC* of the modeling set was 3.0408, the decision coefficient ($R_v^2$) of the verification set was 0.7364, and the *RMSEV* of the verification set was 3.3032.

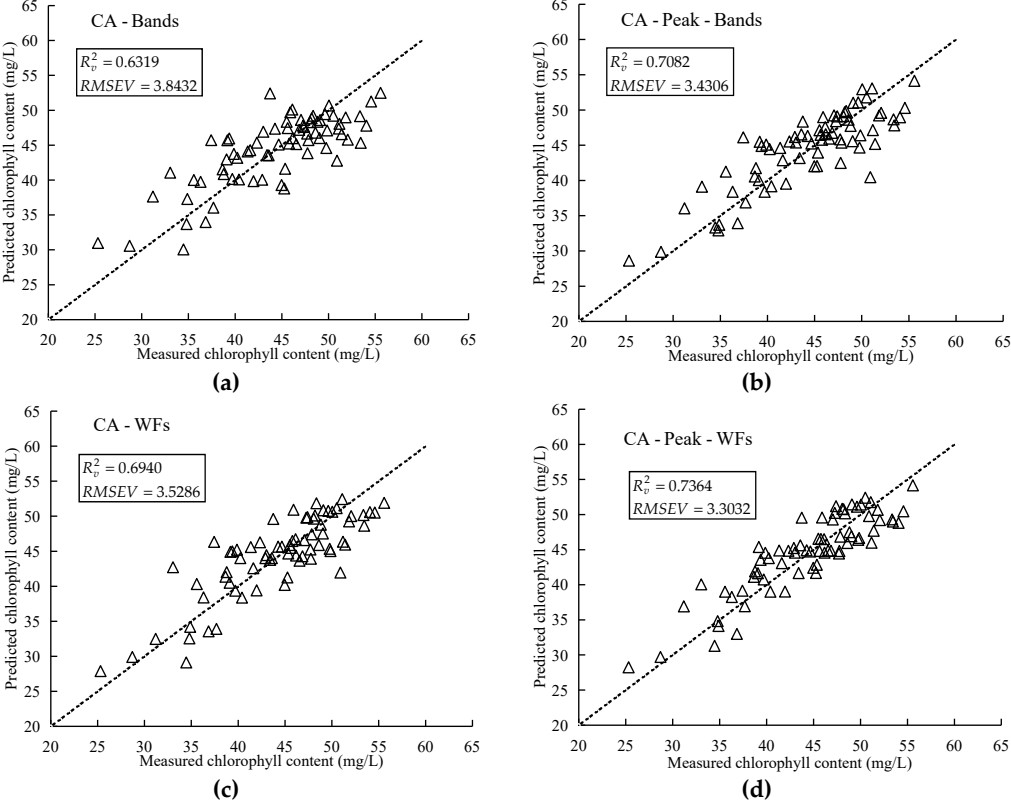

**Figure 8.** Results of the PLSR detection model for the chlorophyll content. (**a**) Results of the maximum correlation coefficient method; (**b**) Results of the local extremum of correlation coefficient method; (**c**) Results of the maximum correlation coefficient method with wavelet energy coefficient; (**d**) Results of the local extremum of correlation coefficient method with wavelet energy coefficient.

**Table 4.** Result statistics of PLSR detection model for the chlorophyll content.

| Characteristic Variable | Number of Variables | Number of Principal Components | Modeling Set | | Verification Set | |
|---|---|---|---|---|---|---|
| | | | $R_c{}^2$ | RMSEC | $R_v{}^2$ | RMSEV |
| CA bands | 15 | 4 | 0.6959 | 3.6214 | 0.6319 | 3.8432 |
| CA peak bands | 12 | 3 | 0.7622 | 3.2015 | 0.7082 | 3.4306 |
| CA-WFs | 50 | 17 | 0.7820 | 3.0661 | 0.6940 | 3.5286 |
| CA peak WFs | 55 | 13 | 0.7856 | 3.0408 | 0.7364 | 3.3032 |

### 3.6. Chlorophyll Distribution

The spectral reflectance of the corn canopy in three growing stages was introduced into the chlorophyll content detection model. The distribution of chlorophyll in the field during the three growing periods was obtained (Figure 9).

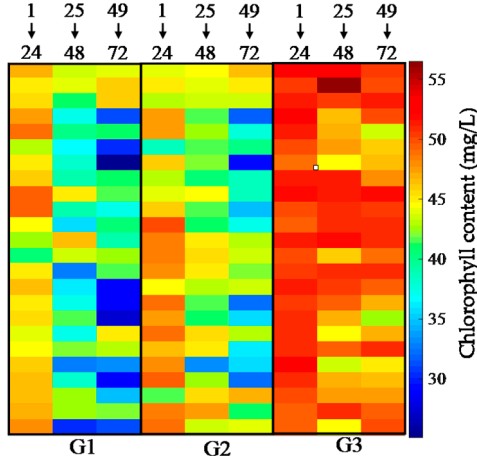

**Figure 9.** Field chlorophyll distribution in three growth stages.

## 4. Discussion

### 4.1. Sensitive Spectral Wavelengths

This study demonstrated that spectral measurements can be used for corn CCC detection. In the range of visible light, the chlorophyll content in the blue and red light regions was positively correlated with spectral reflectance, and reflection peaks were present; in the green light region, the chlorophyll content was negatively correlated with spectral reflectance, and an absorption valley was present [16,51,52]. When the maximum correlation coefficient method was used to filter the sensitive wavelengths, the 15 selected wavelengths were all in the absorption valley of green light, and a serious multicollinearity existed between them. The sensitive wavelengths screened by the local extremum of correlation coefficient method were relatively dispersed and distributed in the visible and near-infrared regions. In the visible light region, the chlorophyll was is characterized by the reflection peak in the red and blue light regions and the absorption valley in the green light region. The extremum characteristic wavelengths in the near-infrared region reflected the composition of other substances [53–55]. The reflectance of the 896 nm band indicates $CH_3$ methyl groups, that in the 976 nm band reflects the moisture content, and that in the 1051 nm band reflects $CH_2$ methylene groups [56–58]. These substances are relative to the canopy structure of the corn crop. Therefore, the characteristic wavelengths in the NIR region will improve the robustness of the chlorophyll detection model. The CA peak bands were more evenly distributed and had less redundant information compared with CA bands.

### 4.2. Continuous Wavelet Analysis

Each wavelet feature contains the information of scale and wavelength position, which corresponds to the state of the generating wavelet function in the process of CWT, namely the scaling factor and the position of shift. The physical meaning of the wavelet feature can be explained by plotting the generating wavelet function corresponding to the wavelet feature that is sensitive to the biochemical parameters. In this study, the Gaussian second derivative was chosen as the generating function of CWT. Each wavelet feature reflects the similarity between the generating wavelet function and the reflectivity spectrum at a specific wavelength position and scale. The absorption characteristics of biochemical parameters at different positions and intensities in the reflectivity spectrum were detected. This situation can be seen as the result of smoothing the spectrum at a particular wavelength and finding the second derivative. The spectral bending degree caused by the different absorption intensities of the biochemical parameters in various bands can be characterized. These two understandings of CWT can be combined to explain the physical meaning of wavelet transform.

The absorption of chlorophyll in the green light band was weaker than that in red and blue light regions. In the reflectivity spectrum, a reflection peak was formed in the green light band. The wavelet features sensitive to chlorophyll were located near the reflection peak of the green light band. Different chlorophyll contents can affect the shape and size of the reflection peak. These changes are easily captured by wavelet features in low scales. A small wavelet feature of medium and high scale near the green light band covered the whole visible band, thus providing the amplitude information of reflectance. The wavelet features at red edge and near-infrared band were stable at the leaf level, thereby indicating that this region is important for chlorophyll monitoring.

Wang et al. used a Mexican hat as the generating wavelet function to obtain the correlation between the wavelet coefficient and the SPAD value of wheat leaves, which can reflect the amount of chlorophyll content. The results showed that the wavelet features sensitive to chlorophyll were located in the red-edge band of 720–740 nm, which was consistent with the research results in References [59–62]. The multiple scattering of light inside the leaves leads to high reflectivity in the near-infrared region, resulting in the rapid increase of spectral reflectance of green vegetation at 680–750 nm, the "red edge" of the area. In the vegetation reflectance spectrum curve, the red edge is one of the most obvious spectral features, and it is an important indicator band used to describe the chlorophyll state of vegetation. The red-edge position (REP) is the wavelength position at which the reflectance of vegetation increases fastest in this interval, and it is also the inflection point of the first derivative of spectral reflectance in this interval. The position of the red edge is an important spectral parameter for detecting the chlorophyll content of vegetation. Liao et al. extracted the wavelet characteristics sensitive to the chlorophyll content of maize leaves in different layers on the basis of the canopy spectrum. The wavelet features sensitive to the chlorophyll content in the upper leaves were distributed in green light and red-edge bands. Meanwhile, the wavelet features sensitive to chlorophyll content in the middle and lower leaves were all located in the red-edge bands, and those in the green light bands disappeared [63]. This phenomenon could have been due to the strong absorption of chlorophyll in the visible range that made it difficult for the green light to penetrate to the middle and lower layers; the visible band of the canopy spectrum mainly contained information regarding chlorophyll in the upper leaves [64]. The comparison of existing literature indicated that the wavelet features at the green light band at the blade level perform better among different datasets. The wavelet features at the red edge perform better among the different datasets at the canopy level.

### 4.3. Sensitive Wavelet Features

The frequency-scale analysis indicated that CA-WFs were mainly distributed on the low frequency scale (1–4), and the other two types were distributed on the mid-frequency scale (5). The wavelength position analysis indicated that CA-WF1 and CA-WF10 were distributed in the blue light region, and CA-WF 3, CA-WF 5, CA-WF 6, CA-WF 8, CA-WF9, and CA-WF11 were distributed in the red light area. These regions are the strong absorption bands of leaf chlorophyll. CA-WF2, CA-WF4,

and CA-WF7 were distributed in the green light regions. These regions are strong reflection bands of leaf chlorophyll. All wavelet features were distributed in the visible light region, which represented the pigment information of the leaves. CA-WFs selected using the maximum correlation coefficient method also had redundant information. For example, the 12 characteristic wavelet coefficients of CA-WF10 were distributed in 472–483 nm, and had serious variable autocorrelation. These autocorrelation problems also existed in CA-WF7, CA-WF8, and CA-WF9.

The frequency-scale analysis indicated that CA peaks WF1–WF4 were distributed in the low-frequency scale (1–4). Meanwhile, CA peaks WF5–WF7 were distributed in the medium-frequency scale (5–7), and CA peak WF8 was located in the high frequency scale (8). Most wavelet features were in the middle- and low-frequency scales, which was consistent with the result of Yao et al [47]. The wavelet features in middle and low frequencies could effectively detect the water content of wheat leaves.

The analysis from the distribution of characteristic wavelet features indicated that all the variables in CA-WFs were distributed in the visible light area and could only reflect the pigment information in the leaves. CA peak WFs were more evenly distributed compared with CA-WFs. The sensitive variable WF4—839 nm in the near-infrared region—reflected the molecular structure of RNHR. WF5—915 nm reflected the $CH_2$ methylene group, and WF5—973 nm showed the moisture information. WF6—985 nm reflected the starch material, WF7—894 nm showed the $CH_3$ methyl group, and WF8—909 nm provided protein information. CA peak WFs comprehensively reflected the material structural information of corn leaves, thereby improving the stability of the chlorophyll detection model.

### 4.4. Chlorophyll Content Detection Model

From the perspective of the method of selecting feature variables, the $R_v^2$ values of the detection models established by using as feature variables the CA bands and CA-WFs were 0.6319 and 0.6940, respectively. The $R_v^2$ values of the detection models established by using as feature variables the CA peak bands and CA peak WFs were 0.7082 and 0.7364, respectively. The distributions of CA bands and CA-WFs were relatively concentrated. A high degree of autocorrelation existed between variables. Meanwhile, CA peak bands and CA peak WFs were evenly distributed and comprehensively reflected information.

Comparing the CA peak bands and the CA peak WFs, under the same variable selection method, the $R_v^2$ of the detection model established using the CA peak WFs (0.7364) was larger than that using CA peak bands (0.7082). The CWT, a process of dimensionality-increasing operation, could dig out the spectral variable information of chlorophyll. The CA peak WFs provided more variable information related to chlorophyll content. Finally, the PLSR model established using CA peak WFs was preferred to detect the chlorophyll content of corn crops.

Comparing the detection models established using spectral reflectance ($R_c^2 = 0.77$) [14] and spectral index ($R_c^2 = 0.70$) [15], the wavelet features ($R_c^2 = 0.7856$) showed better detection capability, which further illustrated that the CWT can deeply mine the information in spectral data.

The model comparison demonstrated that the data after continuous wavelet decomposition can be used to effectively extract valuable information in the spectral reflectance through the dimensionality-increasing operation. In terms of the physical and chemical parameter inversion, the middle- and low-frequency wavelet features highlighted the characteristics of crop pigment and water absorption. In combination with the local extremum of correlation coefficient method, the interference of multicollinearity was eliminated, and the degree of information redundancy was reduced. The detection model established by combining these two methods showed advantages in accuracy and error elimination.

### 4.5. Chlorophyll Distribution in the Field

The distribution of chlorophyll in Figure 9 demonstrates that the chlorophyll concentration in the canopy of plants gradually increased with the advancement of growth period. The field observation

during the field experiment also conformed to this conclusion. The green leaves slowly become dark with the gradual growth of the corn, as shown in Figure 1. The chlorophyll content gradient in the test field remained unchanged over the three growth periods. The figure shows that the chlorophyll contents in the four regions of No. 28–31, 39–48, 52–55, and 63–72 were lower than those in other regions. Some studies have shown a significant positive correlation between the chlorophyll and the nitrogen contents of plants, and chlorophyll can reflect the nitrogen demand of plants to a certain extent [65–68]. Therefore, the chlorophyll detection model based on spectral reflectance can play a guiding role in smart field management and differentiated fertilization.

### 4.6. Future Work

In this study, the advantages of local correlation coefficient extrema in screening feature variables were compared, the ability of CWT to extract weak spectral information was discussed, and a chlorophyll content detection model for corn canopy based on wavelet energy coefficient was constructed. Advanced methods for crop characteristics' detection like computer vision techniques should be involved [69]. In this experiment, hyperspectral images of sample leaves were collected. It is necessary to establish a more accurate and reliable chlorophyll detection model through hyperspectral images. Furthermore, a larger dataset should be used to verify the adaptability of the algorithm.

### 5. Conclusions

In this study, corn canopy spectrum data were collected for three growing stages. First, an S-G filter and SNV correction were applied to the reflectance spectra. Subsequently, the dynamic migration of the canopy spectral characteristics and the chlorophyll content dynamic changes in the three growing periods of G1 to G3 were analyzed. Extraction of the chlorophyll characteristic variables was carried out. Finally, the PLSR detection model of the maize chlorophyll content was established. The conclusions were as follows.

The noise point of the spectral curve was significantly reduced. The scattering effect of the reflection spectrum was significantly reduced after the preprocessing steps of S-G filtering and SNV correction. The reflectance spectrum increased in the 325–400 and 761–970 nm regions as the growth stage advanced and the growth period shifted. The reflectance decreased in the 401–700 and 971–1075 nm regions as the growth stage advanced.

The characteristic variables selected on the basis of the local extrema of correlation coefficients were more evenly distributed compared with the maximum correlation coefficient method. This method weakened the autocorrelation and information redundancy of variables and reflected comprehensive information. The wavelet coefficient obtained by performing CWT on the reflectance spectra was used to efficiently analyze the information of chlorophyll and leaf structure substances in a deep and comprehensive way. The spectral feature extraction method based on CWT highlighted the spectral reflectance features of a specific scale, while suppressing the noncorrelated spectral features and noise of other spectral bands with high flexibility. The proposed method effectively improved the matching accuracy of spectral features.

The highest $R_v^2$ was for the detection model established using the CA peak WFs. The results showed that the CA peak WFs had excellent detection capability for chlorophyll content. CWT combined with the local extremum of the correlation coefficient method is a potentially accurate and efficient strategy for detecting the chlorophyll content of corn crops.

**Author Contributions:** Conceptualization, J.Z. and M.L.; methodology, J.Z.; software, D.G.; validation, L.Q., N.L.; formal analysis, J.Z.; investigation, H.S.; resources, H.S. and Y.Z.; data curation, J.Z.; writing—original draft preparation, J.Z.; writing—review and editing, H.S.; supervision, M.L.; project administration, H.S.; funding acquisition, M.L. All authors have read and agreed to the published version of the manuscript.

**Funding:** The project was supported by the National Key Research and Development Program of China (Grant No. 2016YFD0200600-2016YFD0200602), the National Natural Science Foundation of China (Grant No. 31971785 and 31501219), the National Key Research and Development Program of China (Grant No. 2018YFD0300505-1),

the Fundamental Research Funds for the Central Universities (Grant No. 2020TC036) and the Graduate Training Project of China Agricultural University (JG2019004 and YW2020007).

**Acknowledgments:** The authors wish to thank students in Key Laboratory of Modern Precision Agriculture System Integration Research for the help of Sample processing and chemical analysis. Also, many thanks to Dry Farming Institute of Hebei Academy of Agricultural and Forestry Sciences for providing experimental site. We would like to thank Zizheng Xing, Zhiyong Zhang, Ning Liu, Longsheng Cheng, and Song Li for their help with field data collection.

**Conflicts of Interest:** The authors declare no conflict of interest.

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
