# Peer review of "Detection of Canopy Chlorophyll Content of Corn Based on Continuous Wavelet Transform Analysis"

_remotesensing, doi:10.3390/rs12172741_

Round 1

Reviewer 1 Report

  1. Abstract. Line 31 says "The results of this study can provide guidance for the follow-up work of variable fertilization"; the third aim of the study is to compare the model detection results in step (2) and field variable fertilization data to verify the guiding significance of the model for variable fertilization; and in materials and methods, it is said that the crop had six different fertilization levels. There is no mention about fertilization in the rest of the article. 
  2. English. Check it, specially in lines 62, 136, 276, 419
  3. Materials and Methods. How many samples were collected from each field with different fertilization levels? I guess from figure "7" that the samples are 72*3=216. I would like more clarity about that.
  4. Table 2. Looks like only CA-WF6 is the only one with IrI >0.8
  5. Figures A. Numbering is wrong, there are two figures 7 and its text references must be fixed.
  6. Figures B. Some graphics have no legends.
  7. Discussion A. I would like to see more references to support one of the main statement of the article: the wavelet features sensitive to chlorophyll were located in the red edge band and green light band of 720–740 nm.
  8. Discussion B. Lines 433-441 are results and nothing is discussed in those sentences.
  9. Conclusions. The text should show the conclusions of the study instead of an abstract.
  10. References. Some years are not in bold as the rest of them.

Reviewer 2 Report

Dear Authors,

General comments:

The manuscript evaluates the detection of chlorophyll content in the corn using hyperspectral technology during the different growth stages. The correct application of nitrogen to different crops through the study of chlorophyll content in the leaf has an influence on the improvement of the environment.

Great efforts by the authors!.

The content of the work is easy to read and understand. In the introductory section, it adequately explains the problem of the research and proposes an adequate methodology for its study.

Please consider some comments and suggestions:

Title: The title of your manuscript is concise, specific and relevant.

It is recommended to follow the "Instructions for Authors" for the  experimental and research data to be openly available either by uploading in a cloud file, or by publishing the data and files as supplementary information in this journal.

Section 2.1. Experiments and Materials

Although the location of the test field is not important for the study, it is recommended to geolocate the study area together with the sampling areas and the different levels of fertilization.

Section 2.2. Field Spectrum Data Collection and Chlorophyll Content Measurement

How is the measurement made with the spectrometer on each sampling area? On each sampling areas, three measurements are taken and averaged, what is the area?

What are the characteristics of the UV spectrophotometry used in the laboratory?

Section 3. Results

It’s recommended some clarifications, development and reference on the use of the Savitzky-Golay filter and the normal variable standard (SNV).

Row 273: Wrong spell of the Word Corn (written“Cron”)

Section 3.3.1 Characteristic Wavelength Selection based on the Maximum Correlation Coefficient Method

Row 291. It’s recommended, for a better understanding, to indicate the different frequencies in the blue and red region.

Row 356. Figure 7. It’s recommended to improve the reading of figure 7, using axis labels for a better explanation of the graph.

The use of hyperspectral technology to estimate the chlorophyll content during the growth stages in the corn crop is ok, however, as mentioned in future possible studies, the relationship between chlorophyll content and nitrogen fertilization in the soil is important.

Kind regards,

Reviewer 3 Report

SUMMARY

The paper addresses the research area related to the relationship between canopy chlorophyll content (CCC) and corn canopy spectral reflectance.

It aims to compare the advantages of maximum correlation value and local correlation extreme value in selecting feature variables, to use CWT to decompose the original spectral data and extract the weak information in the spectrum to evaluate the CCC; and to compare the model detection results and field variable fertilization data to verify the guiding significance of the model for variable fertilization.

Authors claim that the detection of the chlorophyll content in field crops for evaluating the growth status and providing guidance for fertilization.

As a general comment, the manuscript is fluent and well structured.

MINOR COMMENT

L287. Please, consider inserting photos (if available) of the corn fields across the three considered growth stages. It would be useful for the readers considering even which reported in L475.

L315-320. Please, consider inserting a table to organize the wavelength intervals and the frequency scale.

Reviewer 4 Report

In this paper, a corn canopy chlorophyll content prediction methodology is proposed. The introduced method uses the CWT transform of the spectral reflectance in order to build a regression model for providing the canopy chlorophyll content.     Specific comments:   1. The paper is well written and organized. 2. In the introduction section, the authors can also discuss advanced methods that use computer vision techniques for visual analyzing the crop characteristics. Recent work in this direction is the following:   Mavridou, E.; Vrochidou, E.; Papakostas, G.A.; Pachidis, T.; Kaburlasos, V.G. Machine Vision Systems in Precision Agriculture for Crop Farming. J. Imaging 2019, 5, 89.   3. Recently, a very similar paper was published by some of the authors of this manuscript, with the following details:   Liu, N.; Zhao, R.; Qiao, L.; Zhang, Y.; Li, M.; Sun, H.; Xing, Z.; Wang, X. Growth Stages Classification of Potato Crop Based on Analysis of Spectral Response and Variables Optimization. Sensors 2020, 20, 3995.   Please the authors provide a detailed discussion about the differences between the above-published work and the methodology proposed in this manuscript.   4. The experiments clearly demonstrate the acceptable performance of the proposed methodology. However, the absence of any comparative study with other similar methodologies reduces the utility of the proposed methodology. The authors should compare their methodology with other techniques under the same experimental protocol.   5. The used data seems to be of small size, only 144 samples for creating the regression model. It would be very constructive if the authors evaluate their methodology for a bigger dataset.    6. From Fig.3 it seems that the task of deciding the corn growth stage from the chlorophyll content is quite non-linear, meaning that it is difficult to distinguish G1 from G2 just using only the chlorophyll content as a feature. Therefore the prediction of the chlorophyll content seems not to be useful for this task.     

Round 2

Reviewer 1 Report

Congratulations.

I enjoyed reading your work.

Best regards

Reviewer 4 Report

All the reviewer's comments have been addressed in a significant degree.
The manuscript can be published in its current form.